

# Plant-exuded chemical signals induce surface attachment of the bacterial pathogen *Pseudomonas syringae*

Megan R. O'Malley[1], Eyram Kpenu[2,3], Scott C. Peck[2,3,4] and Jeffrey C. Anderson[1]

[1] Department of Botany and Plant Pathology, Oregon State University, Corvallis, Oregon, United States of America
[2] Christopher S. Bond Life Sciences Center, University of Missouri, Columbia, Missouri, United States of America
[3] Interdisciplinary Plant Group, University of Missouri, Columbia, Missouri, United States of America
[4] Department of Biochemistry, University of Missouri, Columbia, Missouri, United States of America

Corresponding author
Jeffrey C. Anderson,
anderje2@oregonstate.edu

## ABSTRACT

Many plant pathogenic bacteria suppress host defenses by secreting small molecule toxins or immune-suppressing proteins into host cells, processes that likely require close physical contact between pathogen and host. Yet, in most cases, little is known about whether phytopathogenic bacteria physically attach to host surfaces during infection. Here we report that *Pseudomonas syringae* pv. tomato strain DC3000, a Gram-negative bacterial pathogen of tomato and Arabidopsis, attaches to polystyrene and glass surfaces in response to chemical signals exuded from Arabidopsis seedlings and tomato leaves. We characterized the molecular nature of these attachment-inducing signals and discovered that multiple hydrophilic metabolites found in plant exudates, including citric acid, glutamic acid, and aspartic acid, are potent inducers of surface attachment. These same compounds were previously identified as inducers of *P. syringae* genes encoding a type III secretion system (T3SS), indicating that both attachment and T3SS deployment are induced by the same plant signals. To test if surface attachment and T3SS are regulated by the same signaling pathways, we assessed the attachment phenotypes of several previously characterized DC3000 mutants, and found that the T3SS master regulator HrpL was partially required for maximal levels of surface attachment, whereas the response regulator GacA, a negative regulator of T3SS, negatively regulated DC3000 surface attachment. Together, our data indicate that T3SS deployment and surface attachment by *P. syringae* may be co-regulated by the same host signals during infection, possibly to ensure close contact necessary to facilitate delivery of T3SS effectors into host cells.

## INTRODUCTION

*Pseudomonas syringae* are Gram-negative bacteria capable of infecting a diverse range of plant species. Under environmental conditions that favor disease, *P. syringae* invade through wounds or natural openings in the leaf surface to colonize the intercellular space, or apoplast (*Hirano & Upper, 2000*). Once inside the apoplast, *P. syringae* can proliferate to high levels and ultimately cause visible symptoms including chlorosis and the formation of water-soaked necrotic lesions. *P. syringae* infection can drastically reduce the photosynthetic capacity and productivity of diseased plants (*Lamichhane, Messéan & Morris, 2015*), thereby limiting overall agricultural yield and product quality (*Xin, Kvitko & He, 2018*).

*P. syringae* overcomes host immune responses by deploying a type III secretion system (T3SS), a syringe-like apparatus that delivers proteins termed effectors directly into host cells. Within the host, effector proteins target and suppress immune defense signaling pathways, allowing for pathogen growth in the apoplast. The expression of T3SS-encoding genes is under direct control of HrpL, an alternative sigma factor that directly binds to a *hrp* box motif within the promoters of T3SS-associated genes to activate their expression (*Xiao & Hutcheson, 1994*; *O'Malley & Anderson, 2021*). Although T3SS-associated genes are a primary component of the HrpRS-HrpL regulon, additional T3SS-independent genes have altered expression levels in mutants lacking *hrpL*, suggesting a broader and more complex role for this pathway in regulating gene expression (*Buell et al., 2003*; *Ferreira et al., 2006*; *Lam et al., 2014*).

Genes that encode T3SS components and effectors are not constitutively expressed and must be induced during infection (*Rahme, Mindrinos & Panopoulos, 1992*; *Salmeron & Staskawicz, 1993*). Regulation of T3SS-encoding genes is complex, with distinct inputs from different classes of plant-derived metabolites, as well as from general environmental conditions such as pH. Synthetic media that mimic apoplast conditions, namely an acidic pH, low nitrogen-to-carbon ratio of available nutrients, and the presence of a simple sugar such as fructose, can stimulate expression of T3SS-associated genes (*Huynh, Dahlbeck & Staskawicz, 1989*; *Rahme, Mindrinos & Panopoulos, 1992*; *Salmeron & Staskawicz, 1993*; *O'Malley & Anderson, 2021*). Recently, specific amino acids and organic acids exuded from Arabidopsis seedlings were identified as inducers of T3SS gene expression (*Anderson et al., 2014*), and genes required for T3SS induction by two of these metabolites, aspartic acid and glutamic acid, were identified (*Yan et al., 2020*). Whether these same plant-derived metabolites can also induce T3SS-independent responses in *P. syringae* remains unknown.

Upon entry into the leaf apoplast, *P. syringae* shed their flagella and transition from motile to sessile (*Chinchilla et al., 2006*; *Buscaill et al., 2019*; *Bao et al., 2020*). Electron micrograph imaging of *P. syringae* within the apoplast revealed an apparent close association of bacteria with plant cell surfaces, though the status of these non-motile bacteria, and how they may attach, is not fully understood (*Bestwick, Bennett & Mansfield, 1995*; *Misas-Villamil, Kolodziejek & van der Hoorn, 2011*; *Rufián et al., 2018*). This motility transition may be mediated in part by GacA, a global regulator of virulence known to promote flagellar motility in *P. syringae* and other bacteria (*O'Malley et al., 2020*, *López-*

*Pliego et al., 2021*). However, the status of these non-motile bacteria within the plant host, and the mechanisms by which they may attach, are not fully understood. The syringe-like ultrastructure of the T3SS itself implies that the T3SS may span the plant host cell wall to deliver effectors. The T3SS consists of a ring-like structure that spans both bacterial inner and outer membranes, and an extracellular ~2 μm-long filament termed the pilus likely traverses the plant cell wall, a barrier ranging in thickness from 100 nm to 10 μm (*O'Neill & York, 2003*; *Cornelis, 2006*). Based on similarities between T3SS pilus length and plant cell wall thickness, *P. syringae* likely require close physical contact with the plant cell wall to successfully deliver effectors into host cells, yet mechanisms that may facilitate close physical contact are unknown. Under normal plant growth conditions, the leaf apoplast is an air-filled space with only a thin layer of water associated with the surrounding cell wall (*Sattelmacher, 2001*). As a consequence, close association between *P. syringae* and plant cells may be driven in part by bacterial colonization of this air-water interface. However, *P. syringae* may also actively alter its physiology to physically attach to plant cells to facilitate effector delivery and access cell nutrients.

In this work we report that *P. syringae* attaches to physical surfaces in response to specific metabolites exuded by plant tissues. Surface-inducing activity was present in exudates prepared from Arabidopsis seedlings and tomato leaf tissue. To identify the bioactive compounds, we tested a panel of metabolites present in Arabidopsis exudates and found that several metabolites previously found to induce the T3SS (*Anderson et al., 2014*) also stimulate surface attachment, suggesting co-regulation of these distinct processes in *P. syringae* by the same host signals. Although the molecular nature of this surface attachment is not known, surface attachment was influenced by known genetic regulators of the T3SS. The T3SS regulator HrpL was required for maximal attachment during this response. Conversely, GacA, a negative regulator of T3SS-encoding genes, suppressed surface attachment. Together, our data introduce a model in which T3SS and surface attachment are co-induced by plant signals encountered by *P. syringae* during infection.

## MATERIALS AND METHODS

### Preparation of media, metabolite stocks and crystal violet staining solution

*P. syringae* were routinely cultured on King's B (KB) medium (*King, Ward & Raney, 1954*) solidified with 1.5% agar. For all surface attachment assays, a modified *hrp*-inducing minimal medium (MM) (10 mM $K_2HPO_4/KH_2PO_4$ (pH 6.0), 7.5 mM $(NH_4)_2SO_4$, 3.3 mM $MgCl_2$, 1.7 mM NaCl) was prepared and autoclaved prior to use (*Huynh, Dahlbeck & Staskawicz, 1989*; *Anderson et al., 2014*). For biofilm assays in glass culture tubes, MMR minimal medium (7 mM Na-glutamate, 55 mM mannitol, 1.31 mM $K_2HPO_4$, 2.2 mM $KH_2PO_4$, 0.61 mM $MgSO_4$, 0.34 mM $CaCl_2$, 0.022 mM $FeCl_3$, 0.85 mM NaCl) was prepared as described (*Farias, Olmedilla & Gallegos, 2019*). Autoclaved MM and MMR media were stored at room temperature. Sugar and metabolite stocks were prepared at concentrations of 1 M and 20 mM, respectively, in deionized water, 0.2 μM filter sterilized, and stored at 4 °C. A crystal violet staining solution was prepared by dissolving crystal

violet dye in water to a concentration of 0.05% (w/v). The stain solution was then vacuum filtered through a 0.45 μM membrane to remove any undissolved dye and stored at room temperature.

## Bacterial strains and growth conditions

DC3000 *hrpL⁻* and DC3000 AC811 strains were previously reported (*Zwiesler-Vollick et al., 2002*; *Chatterjee et al., 2003*). A complete list of all strains used in this study is provided in Table S1. *P. syringae* were maintained in 25% glycerol stocks at −80 °C. Bacteria were streaked onto KB agar plates supplemented with antibiotics rifampicin (50 μg/mL), spectinomycin (150 μg/mL), and kanamycin (30 μg/mL) as necessary and grown at room temperature for 2 days prior to use.

## Preparation of plant exudates

*Arabidopsis thaliana* wild-type, *mkp1*, and *mkp1 mpk6* plants, all ecotype Wassilewskija, were previously described (*Anderson et al., 2014*). Arabidopsis seeds were germinated and grown for approximately twelve days on agar plates containing 0.5× Murashige & Skoog (MS) medium (*Murashige & Skoog, 1962*). To prepare exudates, seedlings were removed from agar and immersed in water in a 15 mL conical vial at a density of approximately four seedlings per mL. After overnight incubation, the seedlings were removed and the remaining liquid (exudate) was sterilized using a 0.2 μm syringe filter. To prepare tomato leaf exudates, 0.8 cm² leaf disks were punched from fully-expanded 5-week-old *Solanum lycopersicum* (tomato) cv. Rio Grande leaflets. The isolated leaf disks were floated on water for 24 h at a density of 3 disks/mL. The resulting exudate was sterilized using a 0.2 μm syringe filter. All exudates were either used immediately or stored at −20 °C.

## *P. syringae* surface attachment assays

*P. syringae* were scraped from the surface of KB agar plates and suspended in 1 mL of water. To remove residual KB medium, bacteria were pelleted by centrifugation at $21,000 \times g$ for 1 min, then resuspended in 1 mL of water. This process was repeated twice for a total of three washes. Washed bacteria were adjusted to an optical density at $\lambda = 600$ nm ($OD_{600}$) of 1.0 (~$1 \times 10^9$ bacterial colony forming units/mL), then 50 μL inoculated into the well of 24-well polystyrene assay plates containing 450 μL of MM supplemented with 50 mM of sugar (fructose, sucrose, galactose, mannitol, glucose) and/or 200 μM of citric acid, aspartic acid, 4-hydroxybenzoic acid, glutamic acid, glycine, threonine, valine as indicated. For testing of plant exudates, 20 μL of $OD_{600} = 1.0$ bacteria were inoculated into 100 μL of exudate or water mixed with 100 μL of MM supplemented with or without 100 mM of sugar (fructose, sucrose, galactose, mannitol, glucose) in 96-well polystyrene assay plates. For fructose dose response assays in the presence or absence of seedling exudate, the concentration of fructose in MM was increased up to 250 mM as indicated. After inoculation of bacteria, assay plates were sealed with 3M micropore tape and incubated at room temperature under constant light for 16 h without shaking. To stain surface attached cells, a pipette was used to gently aspirate the supernatant from each well. The assay plate wells were then washed with 1 mL of water prior to staining with 0.5 mL of

0.5% (w/v) crystal violet stain for 15 min at room temperature. After removing the staining solution, stained wells were washed three times with water. Stained wells were then destained by addition of 0.5 mL of 95% ethanol and a Tecan Spark 10M microplate reader was used to measure the Absorbance at $\lambda$ = 562 nm (Abs$_{562}$) of the destaining solution. To rule out that the differences in attachment are due to growth differences between the compared strains, planktonic and attached bacteria were enumerated by serial dilution plating. To collect bacteria for dilution plating, the liquid was removed from an assay well and reserved as the planktonic cell fraction. Attached cells were resuspended by addition of 0.5 mL of MM to each well and scraping the well surface with a pipette tip. The resuspended cell fractions were vortexed for ~10 s to disaggregate cells. Planktonic and attached cell fractions were then serial diluted and spotted onto KB agar plates supplemented with rifampicin. After incubating the agar plates for 1–2 days at room temperature a light microscope was used to count colony forming units (cfus) on plate surfaces.

For attachment assays in glass culture tubes, washed bacteria were inoculated into KB medium, MM or MMR to a final OD$_{600}$ = 1.0 (~1 × 10$^9$) in a total volume of 2.0 mL in sterile borosilicate glass culture tubes. Culture tubes were incubated at room temperature without shaking. After 72 h, the liquid in each culture tube was removed by pipetting, and the interior of each tube gently washed with 3 mL of water, then stained with 2 mL 0.05% CV staining solution. After 15 min, the CV staining solution was removed and tubes were washed three times with water. Tubes were air-dried overnight, then photographed.

## RESULTS

### Plant exudates induce attachment of *P. syringae* to physical surfaces

In previous work we reported that exudates prepared from Arabidopsis seedlings, when supplemented with a simple sugar such as fructose, can induce the expression of T3SS-associated genes in *P. syringae* pv. tomato DC3000 (*Anderson et al., 2014*). During these experiments we routinely cultured DC3000 within the wells of polystyrene assay plates. After culturing DC3000 with seedling exudates, we observed that the inner surface of assay plate wells had become cloudy in appearance, suggesting that bacteria, and/or metabolic products exuded by bacteria, were forming a residue in response to the exudate treatment. To assess if this residue was bacteria physically attaching to the assay plates, we cultured DC3000 in a minimal medium (MM), or MM supplemented with Arabidopsis seedling exudate and/or fructose. After 16 h, we decanted the liquid from assay plate wells and stained the wells with crystal violet, a non-specific stain of bacterial cells routinely used to detect biofilms (*O'Toole & Kolter, 1998*). Significantly higher levels of crystal violet (CV) staining occurred in wells that had contained DC3000 cultured in both fructose and Arabidopsis exudate relative to wells that had contained DC3000 cultured in fructose only (Figs. 1A and 1B). No CV staining was detected in wells that contained DC3000 in MM lacking fructose and exudate (Figs. 1A and 1B). Higher levels of CV staining were detected in wells where DC3000 had been incubated in fructose alone compared to the medium-only control wells, albeit to levels lower than observed with both fructose and exudate (Figs. 1A and 1B). By contrast, no CV staining occurred in wells that contained

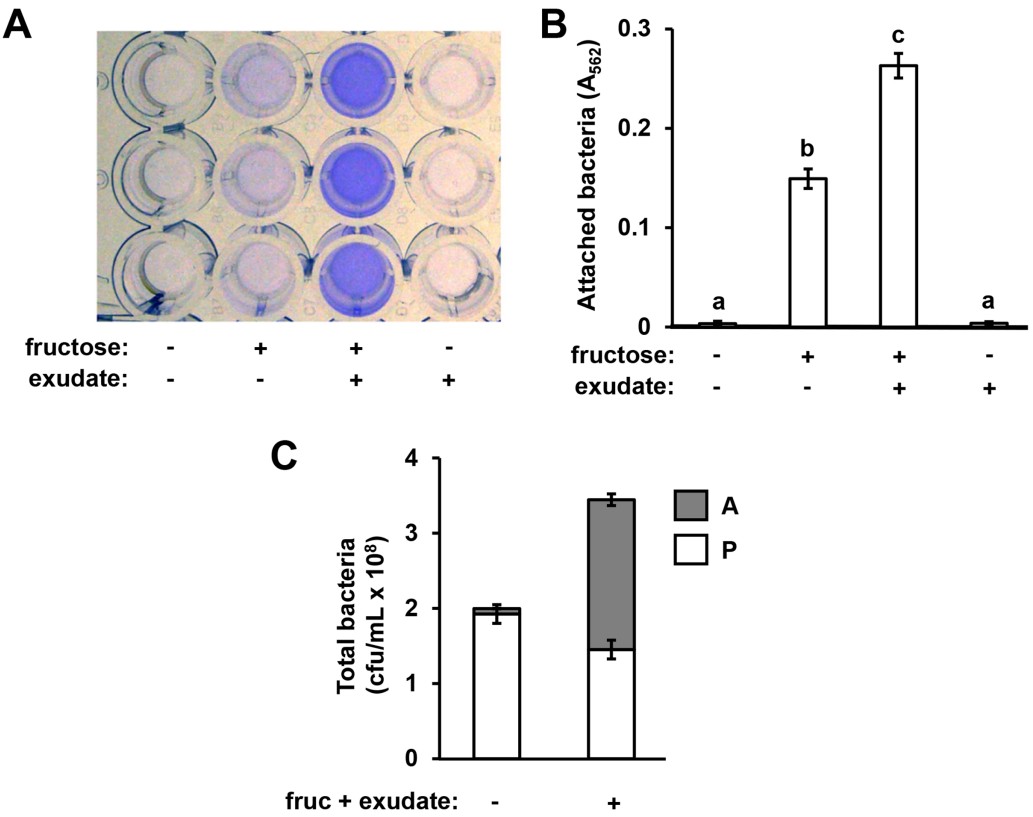

**Figure 1** **Arabidopsis seedling exudates induce *P. syringae* surface attachment.** *P. syringae* DC3000 were cultured in minimal medium (MM) alone, or in MM supplemented with 50 mM fructose and/or Arabidopsis seedling exudate. Cultures were incubated in wells of a 24-well polystyrene microtiter plate without shaking. After 16 h, microtiter plate wells were washed with water to remove unattached bacteria. The assay wells were then either stained with crystal violet (CV), or attached cell fractions were collected to enumerate bacterial levels by serial dilution plating on KB agar. (A) Photograph of plate wells after CV staining and desorption of stain from walls by addition of 95% ethanol. (B) Graphed are means of Absorbance at $\lambda = 562$ nm ($A_{562}$) measurements of wells after CV staining. Lowercase letters are statistical groupings based on multiple pairwise *t*-tests, $p < 0.05$. Error bars are standard error; $n = 6$. (C) Graphed are means of bacterial colony forming units (cfu/mL) from attached (A) and planktonic (P) cell fractions. Error bars are standard error; $n = 4$. Results are representative of at least three independent experiments.

DC3000 and plant exudates in the absence of fructose. We also incubated DC3000 with exudate prepared from tomato leaf tissue and observed similar patterns of CV staining (Fig. S1). To determine if CV staining is due to surface attachment of viable bacteria, we scraped the attached material from the surface of assays wells and plated serial dilutions of these samples, as well as serial dilutions of the supernatant containing planktonic material, on King's B (KB) agar. Surface-attached colony-forming units increased fifteen-fold in wells containing fructose and exudate relative wells containing MM alone, confirming that CV staining is indeed a proxy for levels of surface-attached viable bacteria (Fig. 1C). Total bacterial levels (planktonic plus attached) increased less than two-fold, indicating that increased attachment under these conditions is not due to large changes in overall bacterial growth (Fig. 1C). Together, these observations suggest that signals within both Arabidopsis

and tomato exudates induce surface attachment of DC3000 in a fructose-dependent manner.

A variety of simple sugars, as well as the sugar alcohol mannitol, can induce T3SS gene expression in DC3000 (*Salmeron & Staskawicz, 1993*; *Rahme, Mindrinos & Panopoulos, 1992*). We therefore tested a panel of sugars for surface attachment-inducing activity. Based on CV staining, we detected increased levels of DC3000 attachment in response to all sugars tested, albeit to different levels (Fig. S2). Fructose, galactose, or mannitol combined with exudate elicited the highest levels of attachment, whereas relatively little attachment was observed in response to sucrose or glucose combined with exudate (Fig. S2A). In the absence of seedling exudate, lower but significant levels of surface attachment were observed in MM containing fructose, galactose, or mannitol, whereas MM containing glucose or sucrose did not significantly induce attachment (Fig. S2B). A slight yet significant decrease in attachment was observed in MM containing sucrose relative to MM alone (Fig. S2B). We conclude that all sugars tested, alone or in combination with exudate, can induce attachment and have varying levels of bioactivity.

A diverse mixture of soluble metabolites, including plant-derived sugars, are present in plant exudates (*Anderson et al., 2014*). Because various sugars were able to elicit surface attachment, we reasoned that attachment may be induced by plant-derived sugars present in exudates. To assess this possibility, we incubated DC3000 with or without Arabidopsis exudate in the presence of increasing amounts of fructose. Regardless of the concentration of fructose added (up to 250 mM), the presence of exudate significantly increased the amount of attached bacteria (Fig. S2C). These data suggest that attachment-inducing signals in exudates are not sugars, but distinct signals that function synergistically with exogenously provided sugar to induce attachment.

## Exudates from Arabidopsis *mkp1* stimulate lower levels of *P. syringae* surface attachment

Arabidopsis MITOGEN-ACTIVATED PROTEIN KINASE PHOSPHATASE 1 (MKP1) is a negative regulator of immune-activated MAPK signaling pathways and resistance to bacterial disease (*Jiang et al., 2017*; *Anderson et al., 2011*). Arabidopsis *mkp1* mutants are more resistant to *P. syringae* infection, and this enhanced resistance requires a functional *MAP kinase 6* (*MPK6*) gene. The molecular basis for this enhanced resistance phenotype of Arabidopsis seedlings is decreased exudation of T3SS-inducing signals (*Anderson et al., 2014*). In this regard, *mkp1* seedlings exude lower levels of T3SS-inducing metabolites, whereas exudation of these metabolites from *mkp1 mpk6* double mutant is restored to wild type levels (*Anderson et al., 2014*). To assess if the abundance of surface attachment-inducing signals may be genetically regulated in a similar fashion, we evaluated *mkp1* and *mkp1 mpk6* exudates for their capacity to induce surface attachment. Compared to exudate from wild type plants, exudate from *mkp1* plants stimulated significantly lower levels of surface attachment by DC3000. Conversely, the level of attachment induced by *mkp1 mpk6* exudate was not significantly different from wild type levels (Fig. 2). These data show that,

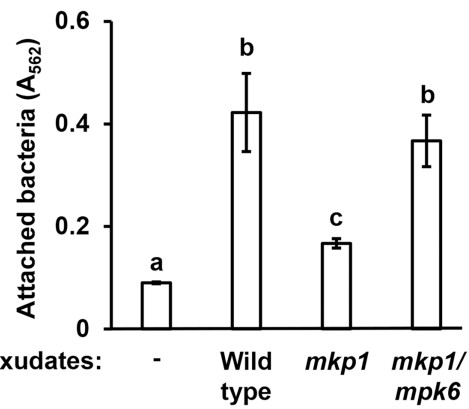

**Figure 2 Exudates from Arabidopsis *mkp1* seedlings elicit lower levels of *P. syringae* surface attachment.** DC3000 were cultured for 16 h in MM containing 50 mM fructose and supplemented with or without exudate from wild type (Ws, ecotype Wassilewskija), *mkp1*, or *mkp1/mpk6* Arabidopsis seedlings as indicated. Graphed are means of Absorbance ($A_{562}$) measurements of crystal violet-stained wells. Small case letters are statistical groupings based on multiple pairwise *t*-tests, $p < 0.05$. Error bars are standard error; $n = 3$. Data are representative of three independent experiments.

similar to T3SS-inducing signals, *MKP1* negatively regulates attachment-inducing signals in exudates in an *MPK6*-dependent manner.

To further characterize the nature of attachment-inducing signal(s), we used chloroform to fractionate exudates from wild type and *mkp1* plants into aqueous and organic phases, then measured levels of surface attachment of DC3000 in response to metabolites extracted into each phase. Surface attachment was induced only by the aqueous fraction of WT and *mkp1* exudates (Fig. S3). These data are consistent with our previous finding that the T3SS-inducing activity of seedling exudate is present only in the aqueous phase after chloroform extraction (*Anderson et al., 2014*), suggesting that T3SS and surface attachment may be induced by the same hydrophilic signal(s).

### *P. syringae* surface attachment is induced by known T3SS-inducing metabolites

In previous work we identified specific T3SS-inducing metabolites present in Arabidopsis seedling exudates and decreased in abundance in *mkp1* exudates (*Anderson et al., 2014*). To assess whether surface attachment is induced by these same signals, we tested three T3SS-inducing metabolites—aspartic acid, glutamic acid, and citric acid—for their ability to induce DC3000 attachment. All three metabolites individually, in combination with fructose, induced significantly higher levels of surface attachment compared to fructose only control wells (Fig. 3A). We also observed that 4-hydroxybenzoic acid, a T3SS-inducing metabolite associated with secondary metabolic pathways, induced attachment, indicating that compounds from diverse metabolic pathways possess attachment-inducing activity (Fig. 3B). Several amino acids present in Arabidopsis exudates that do not induce T3SS genes, namely glycine, valine and threonine (*Anderson et al., 2014*), did not induce attachment (Fig. 3B). These data show a close correlation between T3SS- and attachment-inducing activities of plant-derived metabolites.

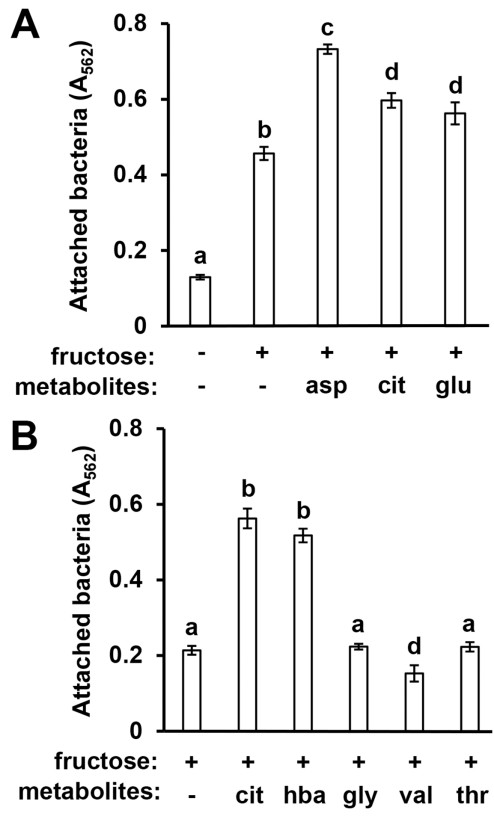

**Figure 3 Known T3SS-inducing metabolites induce *P. syringae* surface attachment.** (A) DC3000 were cultured for 16 h in minimal medium (MM) with or without 50 mM fructose, or MM with 50 mM fructose and 200 µM aspartic acid (asp), citric acid (cit), or glutamic acid (glu). Graphed are means of Absorbance ($A_{562}$) measurements after crystal violet (CV) staining of wells. Statistical groups were determined by multiple pairwise *t*-tests, $p < 0.05$. Error bars are standard error; $n = 4$. (B) DC3000 were cultured for 16 h in MM with 50 mM fructose and 200 µM cit, 4-hydroxybenzoic acid (hba), glycine (gly), valine (val), or threonine (thr). Graphed are means of $A_{562}$ measurements of CV-stained wells. Small case letters are statistical groupings based on multiple pairwise *t*-tests, $p < 0.05$. Error bars are standard error; $n = 4$. Data are representative of at least three independent experiments.

## *P. syringae* surface attachment is influenced by T3SS master regulator HrpL and global regulator GacS-GacA

Because surface attachment and type III secretion are induced by the same metabolites, we next tested whether these responses are dependent on the same signaling pathways in *P. syringae*. We first tested attachment of a DC3000 *hrpL*⁻ insertion mutant and its wild type parental strain carrying either a complementing plasmid (pVSP61::*hrpL*) or an empty vector control (*Zwiesler-Vollick et al., 2002*; *Sreedharan et al., 2006*). Surface attachment of DC3000 *hrpL*⁻ in response to fructose and citric acid was partially attenuated, exhibiting a ~40% reduction in attachment relative to wild type DC3000 (Fig. 4). Introduction of pVSP61::*hrpL* into DC3000 *hrpL*⁻ restored metabolite-induced attachment to wild type levels (Fig. 4). These results indicated that the observed surface attachment is dependent on both *hrpL*-dependent and -independent processes in DC3000.

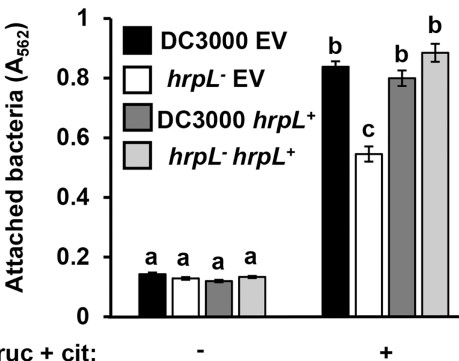

**Figure 4** *P. syringae* **surface attachment is partially dependent on T3SS master regulator HrpL.** DC3000 and DC3000 $hrpL^-$, harboring either a pVSP61::$hrpL$ complementing plasmid ($hrpL^+$) or pVSP61 empty vector (EV), were incubated in minimal medium (MM) supplemented with 50 mM fructose and 200 μM citric acid for 16 h. Graphed are means of Absorbance ($A_{562}$) measurements of crystal violet-stained microtiter plate wells. Lowercase letters denote statistical groups based on ANOVA with multiple pairwise $t$-test comparisons and Tukey's *post-hoc* HSD test, $p < 0.05$. Error bars are standard error; $n = 3$. Data are representative of three independent experiments.

GacS-GacA is a two-component system that regulates the expression of various virulence traits in pseudomonads (*Heeb & Haas, 2001*), including biofilm formation in some species (*Parkins, Ceri & Storey, 2001*; *Li et al., 2015*). To investigate whether GacS-GacA also regulates surface attachment in DC3000, we measured attachment of AC811, a Tn5 insertion mutant of *gacA* (*O'Malley et al., 2019*; *Chatterjee et al., 2003*), in response to fructose and plant metabolites. We observed significantly higher levels of surface attachment of the AC811 mutant in response to fructose and citric acid relative to its wild type DC3000 parental strain (Fig. 5), indicating that surface attachment may be negatively regulated by GacA. Accordingly, levels of surface attachment were suppressed to sub-wild type levels in a complemented strain of AC811 expressing *gacA* under its native promoter (AC811 $gacA^+$). The repression of surface attachment below wild type levels in both DC3000 $gacA^+$ and AC811 $gacA^+$ is likely due to *gacA* overexpression in these complemented strains, as previously described (*O'Malley et al., 2020*). We conclude from these results that GacA functions as a negative regulator of surface attachment by DC3000 in response to fructose and citric acid.

## T3SS-inducing metabolites enhance *P. syringae* attachment at the air-liquid interface in static cultures

We also assessed whether T3SS-inducing metabolites induce surface biofilm formation at the air-liquid interface, a phenomenon previously described in DC3000 (*Farias, Olmedilla & Gallegos, 2019*). To investigate, we cultured DC3000 under static conditions in borosilicate glass culture tubes containing rich KB medium, or containing MM with or without fructose and citric acid. We also cultured DC3000 in MMR, a defined minimal medium previously used to assess *P. syringae* biofilm dynamics (*Farias, Olmedilla & Gallegos, 2019*; *Robertsen et al., 1981*). MMR differs from MM by addition of mannitol and sodium glutamate, and contains higher levels of iron and calcium (*Farias, Olmedilla &*

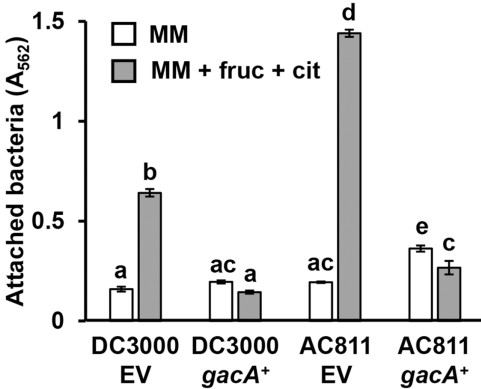

**Figure 5** *P. syringae* **surface attachment is repressed by the global response regulator GacA.** DC3000 and DC3000 *gacA*::Tn5 (strain AC811) carrying either pBBR1-MCS1::*gacA* (*gacA*[+]) or pBBR1-MCS1 empty vector (EV) were incubated in minimal medium (MM) supplemented with 50 mM fructose and 200 μM citric acid for 16 h. Graphed are means of Absorbance ($A_{562}$) measurements of crystal violet-stained microtiter plate wells. Lowercase letters denote statistical groups based on ANOVA with multiple pairwise *t*-test comparisons and Tukey's *post-hoc* HSD test, $p < 0.05$. Error bars are standard error; $n = 3$. Data are representative of three independent experiments.

*Gallegos, 2019*). To assess bacterial attachment, we visually observed culture tubes for 72 h, followed by CV staining of culture tube surfaces. A pellicle, or floating biofilm formed at the air–liquid interface, was visible on the surface of DC3000 in KB as early as 48 h post-inoculation, whereas no pellicles were ever visible in tubes containing DC3000 in MMR or in MM with or without fructose and citric acid (Fig. 6A). These results were distinct from those reported previously, where pellicle formation was observed in MMR, suggesting potential differences in experimental conditions. Upon removal of media and CV staining of culture tubes, we observed only diffuse CV staining on the walls of tubes that contained DC3000 in MM (Fig. 6B). A slight yet more defined ring of CV staining at the air-liquid interface was apparent on the walls of tubes that contained DC3000 in KB or MMR (Fig. 6B). In contrast, a dense ring of CV staining at air-liquid interface was observed in tubes that contained DC3000 in MM supplemented with fructose and citric acid (Fig. 6B).

## DISCUSSION

In this work, we discovered that specific plant-derived metabolites induce *P. syringae* to attach to surfaces. We established that surface attachment is a genetically-encoded response in DC3000, with T3SS master regulator HrpL necessary for maximal attachment, and the global response regulator GacA functioning to suppress attachment. These results support a model in which surface attachment is a process co-activated by T3SS-inducing plant metabolites, possibly as a means to ensure close contact with host cells necessary for T3SS deployment during infection.

While our results demonstrate that *P. syringae* is capable of physical attachment to surfaces, the mechanism(s) of adhesion remains unknown. Among bacterial pathogens, a variety of extracellular structures mediate attachment to surfaces, and attachment processes can be reversible or irreversible. The gall-forming plant pathogen *Agrobacterium tumefaciens* forms reversible attachments to host surfaces through pili, which are multiple

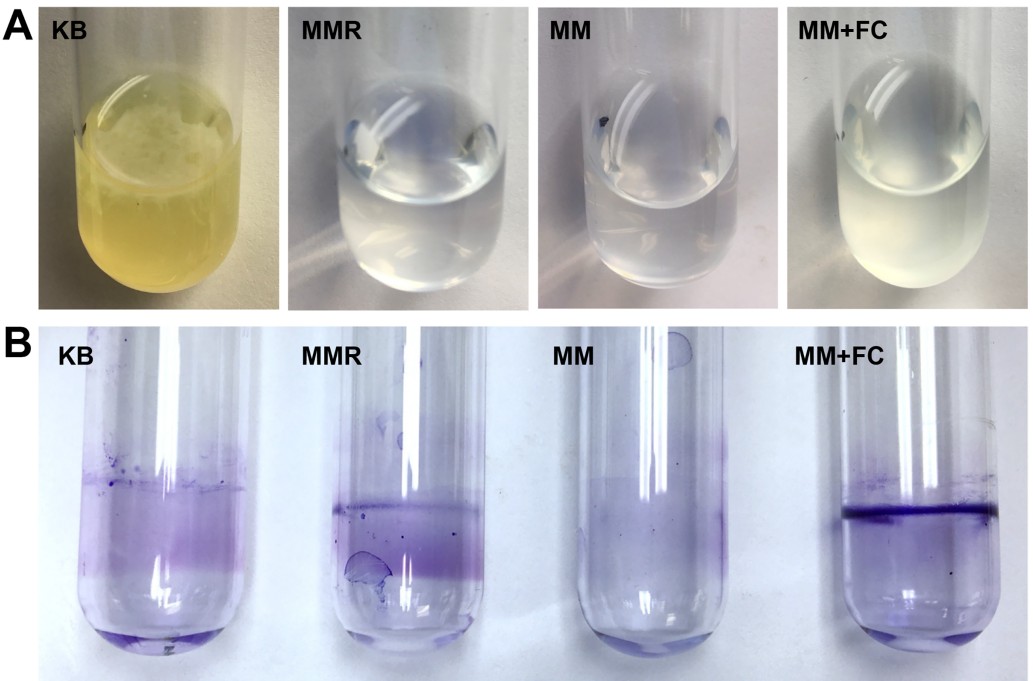

**Figure 6 T3SS-inducing metabolites enhance surface attachment by _P. syringae_ at the air-liquid interface in glass culture tubes.** _P. syringae_ DC3000 were incubated in borosilicate glass tubes in King's B (KB) rich medium, MMR medium, or minimal medium (MM) with or without 50 mM fructose and 200 μM citric acid. Tubes were maintained at room temperature without shaking for 72 h before aspirating the cultures and staining the culture tube surfaces with crystal violet (CV). Pictured are culture tubes after CV staining, with visible rings of bacterial attachment at the level of the air-water interface. Results are representative of three independent experiments.

distinct types of hair-like extracellular appendages (_Wang, Haitjema & Fuqua, 2014_). Similarly, pili mediate initial contact of the opportunistic animal and plant pathogen _Pseudomonas aeruginosa_ with host epithelial cell surfaces (_Bucior, Pielage & Engel, 2012_). Structural components of flagella also contribute to reversible surface association by _P. aeruginosa_ (_Merritt et al., 2007_; _Schniederberend et al., 2019_). Following initial reversible surface contact, bacterial cells often undergo phenotypic differentiation and become irreversibly attached (_Sauer et al., 2002_; _Hinsa et al., 2003_). These cells become anchored to surfaces through attachment factors such as extracellular polysaccharides (EPS) and/or cell-surface proteins such as adhesins or carbohydrate-binding lectins, providing strong surface adhesion. Numerous potential attachment factors are genetically encoded in _P. syringae_, including various pili and extracellular adhesins (_Buell et al., 2003_), though their roles in virulence remain unknown. _P. syringae_ has been observed to form biofilms, multicellular communities in which bacteria are embedded in a self-produced matrix comprised of EPS or other polymeric substances. When grown in rich media, _P. syringae_ forms biofilms consisting of EPS such as alginate and levan (_Laue et al., 2006_), as well as pellicles containing both alginate and cellulose (_Farias, Olmedilla & Gallegos, 2019_). Less is known about biofilm formation during infection, though the production of various EPS is required for epiphytic fitness and/or full virulence of select _P. syringae_ strains on host

plants, suggesting a role for biofilm formation in host infection (*Yu et al., 1999*; *Arrebola et al., 2015*; *Helmann, Deutschbauer & Lindow, 2019*; *Heredia-Ponce et al., 2020*). Further analysis of the genetic basis of *P. syringae* attachment observed in this study will be necessary to determine the physical factor(s) that promote the surface association.

Our analysis of the AC811 *gacA*::Tn5 strain indicates that surface attachment by *P. syringae* may be negatively regulated by the GacS-GacA two component system. Recently we reported that GacA also functions as a negative regulator of T3SS and a positive regulator of flagellar motility (*O'Malley et al., 2019*; *O'Malley et al., 2020*). Based on these previous findings, we hypothesized that GacS-GacA signaling is activated on the leaf surface, where it functions to suppress production of the T3SS and promote flagellar motility for bacteria to gain access to the leaf interior through stomata or other openings (*O'Malley et al., 2020*). Suppression of the T3SS by GacA is also hypothesized to optimize fitness of epiphytic bacteria, given that the T3SS is critical for *P. syringae* survival within the apoplast but is largely dispensable on the leaf surface (*Helmann, Deutschbauer & Lindow, 2019*). Based on the data presented here, we propose that *P. syringae* surface attachment may be similarly repressed on the leaf surface by activated GacS-GacA, and potentially de-repressed within the apoplast due to inactivated GacS-GacA. Analysis of the activation of GacS-GacA within the host plant environment will be necessary to fully evaluate this model.

Upon apoplast colonization, *P. syringae* shed their flagella and undergo a motile-to-sessile transition (*Chinchilla et al., 2006*; *Yu et al., 2013*; *Helmann, Deutschbauer & Lindow, 2019*; *Bao et al., 2020*; *Buscaill et al., 2019*). As such, it is plausible that *P. syringae* surface attachment may occur coincident with with this loss in motility, potentially as a means to facilitate T3SS deployment. Our observations that *P. syringae* attachment is induced by metabolic signals that are highly abundant in the leaf apoplast, such as glutamic acid and aspartic acid, lends further support to this hypothesis (*Rico & Preston, 2008*; *Kumar et al., 2017*). The extended length of the *P. syringae* pilus is hypothesized to facilitate delivery of T3SS effectors across the plant cell wall, which can be up to 10 nm in thickness (*O'Neill & York, 2003*). Although the *P. syringae* T3SS pilus can extend several μM in length, significantly longer than the ~50–600 nm T3SS filament produced by animal pathogenic γ-proteobacteria (*Cornelis, 2006*), close association with the plant cell wall is likely required for the T3SS pilus to contact the host plasma membrane and deliver effectors. Physiochemical effects of bacterial association with plant cell walls, such as local changes in osmolarity and pH in association with surfaces *vs.* bulk liquid, may additionally influence effector delivery and pathogen dynamics during infection (*Hong & Brown, 2010*; *Kimkes & Heinemann, 2020*).

Multiple T3SS-inducing signals, including aspartic acid and citric acid (*Anderson et al., 2014*), as well as a simple sugars, increased surface attachment of *P. syringae*. Conversely, glycine, valine, and threonine, compounds that were previously identified as non-inducers of T3SS (*Anderson et al., 2014*), failed to elicit surface attachment. This apparent close correlation between induction of T3SS and surface attachment, as well as partial requirement of T3SS regulator HrpL for surface attachment, suggests that both responses may be partially regulated by the same signaling pathways in *P. syringae*. In various

pathogens, nonmotile or surface-associated cells exhibit increased T3SS expression and virulence, suggesting that co-regulation of attachment and T3SS may be a common virulence strategy (*Soscia et al., 2007*; *Tamayo, Patimalla & Camilli, 2010*). The T3SS translocon is required for initial adhesion and bacterial aggregate formation on host cell surfaces by various pathogens, including *P. aeruginosa* and the biofilm-forming plant pathogen *Xanthomonas citri* (*Hernandes et al., 2013*; *Santos et al., 2019*; *Tran et al., 2014*; *Zimaro et al., 2014*). These studies indicate that the T3SS translocon itself may function as a surface adhesin, though the underlying mechanism(s) have not been established. Beyond the potential role of the T3SS itself in mediating surface interaction, various T3SS-independent genes are co-regulated by HrpL and may impact attachment (*Lam et al., 2014*). How T3SS deployment and surface attachment may cooperatively function during host infection remains to be determined.

## CONCLUSIONS

This study demonstrates that *P. syringae* pv. tomato DC3000 attaches to physical surfaces in the presence of host plant exudates. Plant metabolic signals that induce type III secretion by *P. syringae*, including citric acid and aspartic acid, were identified as inducers of surface attachment. Our findings demonstrate that surface attachment and T3SS gene expression by *P. syringae* are co-activated by the same plant metabolites, suggesting that these two processes may be co-regulated during infection.

### Funding

This work was supported by USDA National Institute of Food and Agriculture Predoctoral Fellowship 2020-67034-31746 awarded to Megan R. O'Malley, National Science Foundation grants IOS-1456256 and IOS-1953509 awarded to Scott C. Peck, and National Science Foundation grant IOS-1557694 awarded to Jeffrey C. Anderson. The funders had no role in study design, data collection and analysis, decision to publish, or preparation of the manuscript.

### Grant Disclosures

The following grant information was disclosed by the authors:
USDA National Institute of Food and Agriculture Predoctoral Fellowship: 2020-67034-31746.
National Science Foundation: IOS-1456256 and IOS-1953509.
National Science Foundation: IOS-1557694.

### Competing Interests

The authors declare that they have no competing interests.

## Author Contributions

- Megan R. O'Malley conceived and designed the experiments, performed the experiments, analyzed the data, prepared figures and/or tables, authored or reviewed drafts of the article, and approved the final draft.
- Eyram Kpenu performed the experiments, analyzed the data, prepared figures and/or tables, and approved the final draft.
- Scott C. Peck conceived and designed the experiments, analyzed the data, prepared figures and/or tables, authored or reviewed drafts of the article, and approved the final draft.
- Jeffrey C. Anderson conceived and designed the experiments, performed the experiments, analyzed the data, prepared figures and/or tables, authored or reviewed drafts of the article, and approved the final draft.

## Data Availability

The raw data are available in the Supplemental Files.

## Supplemental Information

Supplemental information for this article can be found online at http://dx.doi.org/10.7717/peerj.14862#supplemental-information.

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
