# Peer review of "Plant-exuded chemical signals induce surface attachment of the bacterial pathogen Pseudomonas syringae"

_PeerJ, doi:10.7717/peerj.14862_

## Round 0.1 · original submission · Major Revisions

Thank you for considering PeerJ for your manuscript submission. Taking into account the comments of the three reviewers, I coincide with their opinions. While the referees do acknowledge that the manuscript has merit, the clear consensus is that modifications are needed. It was marked several topics by the reviewers should be addressed to improve the technical quality of your manuscript before submission of a revised version of your manuscript.

Therefore, I consider that the manuscript can be re-submitted as a revised version wherein provided that the recommendations are addressed to reinforce the biological phenomenon studied. Also, you should answer all questions and items requested point by point.

Thank you for your progress, and we look forward.

·

Basic reporting

The introduction is generally consistent with the objectives and results of the work. However, the information presented about the GacS/A regulation system is poor (lines 128-29) compared to what was discussed for the Hrp system (lines 79-88).

The flagellum's role in converting motile to sessile cells is addressed (lines 102-103). This role is essential in the colonization process of the plant. It is known that in several species of Pseudomonas, the GacS/A system controls flagellum synthesis and thus this motility transition. It would be desirable to address this issue in the introduction and consolidate it in the discussion section.

To clarify that the differences observed in Figure 1 are not due to growth differences among the compared strains, I suggest converting Figures S2 and S2B into Figures 1B and 1C.

Explain why the effect of the mutation in MKP6 reverses the impact of the mutation in MKP1 (line 255)

The description of the GacS/A system could be in the introduction section (lines 308-309)

Experimental design

In the description of surface attachment assays (line 181); it is not clear why the planktonic cells and the attached cells are quantified, it is suggested to specify the reason for carrying out this procedure (to rule out that the differences in the attached are due to differences in growth between the compared strains)

Validity of the findings

The relationship between aerial biofilm formation and attachment to solid surfaces is not clearly established. What role could this air-liquid biofilm play in the interaction with the plant? And, what would be the role of the attachment inducers reported in this work? It is suggested to address this issue in the discussion section.

It is reported that orthologs of GacS (BarA) are activated by the presence of ionized organic acids (which depends on the pH of the medium). Interestingly, organic acids are also reported in this work as possible signals of the process. However, the input domains of BarA from E. coli and GacS from P. aeruginosa show differences that indicate they could respond to different signals (DOI: 10.1016/j.jbc.2021.101383). On the other hand, pH is among the factors mentioned that could influence the promotion of plant-bacteria interaction, which could be related to the ionization state of a possible signal molecule. This issue would be an interesting topic to discuss in the corresponding section.

Additional comments

I consider that the work can be enriched by addressing some points listed in the review. Once the comments have been addressed, I consider that the new version could be accepted for publication.

Reviewer 2 ·

Basic reporting

The used English is clear, unambiguous, professional throughout.
The intro & background show context.
The literature is well referenced & relevant.
The structure is conforms to PeerJ standards,
Figures are relevant, high quality, well labeled & described. But:
To appropriately compare panels A and B of figure S2 (Attached and plantonik cells) please use the same scale units in Y axes, (% or log) of the graphs. Figure 6 is not necessary.
Statistical grouping in figure 3B must be checked.
The raw data was supplied.
The submission may be considered self-contained.

Experimental design

The manuscript provide original primary research within Aims and Scope of the journal.
The research question well defined, relevant & meaningful. It is stated how research fills an identified knowledge gap.
The investigation is rigorous performed to a high technical & ethical standard.
Methods described must be improved with sufficient detail & information to be replicate.

Validity of the findings

All underlying data have been provided; they are robust, statistically sound, & controlled.
The conclusions are well stated, linked to original research question & limited to supporting results.

Additional comments

I write in reference to the manuscript “Plant-exuded chemical signals induce surface attachment of the bacterial pathogen Pseudomonas syringae” by O'Malley and co-workers. The title is very descriptive of the manuscript content.
In a previous work, the authors identified different plant metabolites that induce the expression of genes encoding a type III secretion system (T3SS) of P. syringae. In the present work, the authors tested the effect of crude Arabidopsis radical exudates as well as the metabolites previously shown effect on genes encoding T3SS system, they showed that these compounds also induce surface attachment of P. syringae and discussed the implications of this finding. In this sense, the authors found a knowledge gap and worked on it. The knowledge gap being filled is explicitly stated in the introduction and discussion sections.
The manuscript is well written and the conclusions were carefully drafted to not exceed the support of results. Although there are some issues that need to be addressed before the manuscript can be published.
My principal concern is that the concentrations of metabolites used in this study seem very high, it means that they are not physiological concentrations. The significance of the results must be discussed in relation to this point.
Other concerns:
Line 140. Please briefly describe the MMR medium as you did with MM medium.
Line 155. Preparation of plant exudates. Please provide the Arabidopsis ecotype and an appropriate reference. Also mention the different employed Arabidopsis mutant lines and provide the references where they are described. Please mention the Arabidopsis seeds provider as you did with P. syringae strains.
Line 166 P. syringae surface attachment assays. Please describe the use of plant exudates in surface attachment assays.
Line 184. Please indicate how the biofilm was disaggregated.
Lines 122 to 126. Figure S2. To appropriately compare panels A and B (Attached and plantonik cells) please use the same scale units in Y axes, (% or log) of the graphs.
Line 288. Did pH play a role in the decrement of attachment to 500 and 100 µM? Was the pH controlled? Can the authors provide the pH of the medium?
Line 299. pH could also play a role also in the interference of citric acid with the ability of DC3000 to respond to 4-hydroxybenzoic acid. Please comment on this possibility.
Line 397. The concentrations used in this study seem very high. Please provide information (and the correspondent references) about the concentrations of metabolites as glutamic acid, aspartic acid, citric acid, and (specially) fructose in the apoplast of plants as Arabidopsis, and please discuss the results' relevance in reference to these information.
Figure 3B. Please check the statistical grupping. As is mentioned in the text (lines 280-281) Panel B do not show differences between control and gly or thr, but they are qualified with different letters in the graph.
I thank you for providing the raw data, however your supplemental files need more descriptive identifiers to be useful to future readers. Please add titles to the data and notes to identify the experiments with whom they correspond.

Reviewer 3 ·

Basic reporting

This paper reports the identification of metabolites present in plant exudates that induce surface adhesion in P. syringae, a plant pathogenic bacterium. The major question explored here has relevance to understanding how bacterial cells change after entering the leaf apoplast from being motile to non-motile sessile cells capable of making a contact with plant cells long enough for bacterial effector proteins to be secreted into plant cells. While this paper does not investigate or make claims about whether the surface adhesion is important to plant interactions, it does show that surface adhesion behavior and virulence gene expression are both activated by the same plant metabolites and that there is a partial dependence on the master virulence regulator (HrpL) for surface adhesion.

Overall, this paper is very well written and a pleasure to read. The story is laid out very logically, I have no issues with any aspect of the presentation. The writing is clear and concise, the figures are of sufficient quality, background is up-to-date and well referenced.

Experimental design

This study followed from an observation of P. syringae cells that appeared to adhere more to the surface of culture wells when they were grown under the same conditions as those that induce expression of the bacterium’s virulence systems. From there the authors lay out a series of straightforward experiments to determine that these cells are viable and resembled biofilm-like surface adhesion properties (although never mentioning the word "biofilm"). After establishing that correlation the authors tested several predictions that were based off their previous work on virulence system induction. All the experiments described are straightforward and well controlled.

Validity of the findings

In general, the results are valid, as would typically follow from well-designed experiments. I found one small issue in Figure 3, the variance for +fructose alone appears significantly different in 3A and 3B. I’ll add that this paper has two noticeable limitations. One is that the mechanism of surface adhesion was not explored. The other is that the biological significance of surface adhesion with regard to virulence was not tested. However, even with those limitations, this is solid work and seems to meet PeerJ’s criteria.

---

## Round 0.2 · accepted · Accept

The authors have satisfactorily complied with the suggested corrections of reviewers, the manuscript was improved and addressed as recommended. I consider that the manuscript in the current version can be accepted for publication.

·

Basic reporting

The authors complied with the recommendations satisfactorily.

Experimental design

The authors complied with the recommendations satisfactorily.

Validity of the findings

The authors complied with the recommendations satisfactorily.

Additional comments

The authors have satisfactorily complied with the suggested corrections, so I consider that the article in this version can now be accepted.

Reviewer 2 ·

Basic reporting

I consider that the work has been enriched by addressing the points listed by the reviewers and now could be accepted for publication.

Experimental design

I consider that my concerns have been attended to.

Validity of the findings

I consider that the work has been enriched by addressing the points listed by the reviewers and now could be accepted for publication.

Additional comments

No more comments.